# Protocol for a single-centre mixed-method pre–post single-arm feasibility trial of a culturally appropriate 6-week pulmonary rehabilitation programme among adults with functionally limiting chronic respiratory diseases in Malawi

Fanuel Meckson Bickton [1,2] Talumba Mankhokwe,[3] Rebecca Nightingale,[1,4] Cashon Fombe,[3] Martha Mitengo,[3] Langsfield Mwahimba,[3] Wilfred Lipita,[3] Laura Wilde [5,6] Ilaria Pina,[5,6] Zainab K Yusuf [5,6] Zahira Ahmed,[5,6] Martin Kamponda,[7] Felix Limbani,[1] Harriet Shannon,[2] Enock Chisati,[8] Andy Barton,[5] Robert C Free,[5] Michael Steiner,[5,6] Jesse A Matheson,[9] Adrian Manise,[10] Sally J Singh,[5,6] Jamie Rylance,[1,4] Mark Orme [5,6]

JR and MO are joint senior authors.

For numbered affiliations see end of article.

**Correspondence to**
Fanuel Meckson Bickton;
fbickton@mlw.mw

## ABSTRACT

**Introduction** Malawi has a substantial burden of chronic respiratory diseases (CRDs) which cause significant morbidity and loss of economic productivity, affecting patients, families and health systems. Pulmonary rehabilitation (PR) is a highly recommended non-pharmacological intervention in the clinical management of people with CRDs. However, Malawi lacks published evidence on the implementation of PR for people with CRDs. This trial will test the feasibility and acceptability of implementing a culturally appropriate hospital-based PR programme among adults with functionally limiting CRDs at Queen Elizabeth Central Hospital in Blantyre, Malawi.

**Methods and analysis** This is a single-centre mixed-methods pre–post single-arm feasibility trial. Ten patients aged ≥18 years, with a spirometry confirmed diagnosis of a CRD and breathlessness of ≥2 on the modified Medical Research Council dyspnoea scale, will be consecutively recruited. Their baseline lung function, exercise tolerance and health status will be assessed; including spirometry, Incremental Shuttle Walk Test and Chronic Obstructive Pulmonary Disease Assessment Test, respectively. Pretrial semistructured in-depth interviews will explore their experiences of living with CRD and potential enablers and barriers to their PR uptake. Along with international PR guidelines, these data will inform culturally appropriate delivery of PR. We initially propose a 6-week, twice-weekly, supervised centre-based PR programme, with an additional weekly home-based non-supervised session. Using combination of researcher observation, interaction with the participants, field notes and informal interviews with the participants, we will assess the feasibility of running the programme in the following areas: participants' recruitment, retention, engagement and protocol adherence. Following programme completion (after 6 weeks), repeat assessments of lung function,

### Strengths and limitations of this study

► Pre-trial interviews with participants will result in a 'culturally adapted' intervention.
► Post-trial interviews will result in a refined and more acceptable intervention.
► The design of this trial will inform the design of a randomised controlled trial.
► Small sample size and single-centredness will limit the external validity of this trial.
► The cohort design of this trial will not ascertain a cause and effect relationship.

exercise tolerance and health status will be conducted. Quantitative changes in clinical outcomes will be described in relation to published minimal clinically important differences. Post-trial semistructured interviews will capture participants' perceived impact of the PR programme on their quality of life, enablers, and barriers to fully engaging with the programme, and allow iteration of its design.

**Ethics and dissemination** Ethical approval for this trial was obtained from University of Malawi College of Medicine Research and Ethics Committee (COMREC), Blantyre, Malawi (protocol number: P.07/19/2752) and University of Leicester Research Ethics Committee, Leicester, UK (ethics reference: 31574). The results of the trial will be disseminated through oral presentations at local and international scientific conferences or seminars and publication in a peer-reviewed journal. We will also engage the participants who complete the PR trial and the Science Communication Department at Malawi-Liverpool-Wellcome Trust Clinical Research Programme to organise community outreach activities within Blantyre to educate communities about CRDs and PR. We will also broadcast

our trial results through national radio station programmes such as the weekly "Thanzi la Onse" (Health of All) programme by Times Radio Malawi. We will formally present our trial results to Blantyre District Health Office and Malawi Ministry of Health.

**Trial registration number** ISRCTN13836793.

## INTRODUCTION

Low-income and middle-income countries (LMICs) bear a disproportionately high burden of the global morbidity and mortality caused by chronic respiratory diseases (CRDs), including asthma, chronic obstructive pulmonary disease (COPD), bronchiectasis and post-tuberculosis lung disease (p-TBLD).[1] Known risk factors include tobacco smoking, outdoor air pollution, household smoke exposure, occupational dust exposure and pulmonary tuberculosis.[2] Significant chronic morbidity and loss of economic productivity are related to CRDs, which burden patients, families and health systems.[3]

A recent meta-analysis of Malawi data identified a high burden of chronic respiratory symptoms and abnormal spirometry data (particularly low forced vital capacity) in both children and adults.[4] Notably, even successful tuberculosis (TB) treatment is still frequently followed by long-term cough and shortness of breath, leading to impaired quality of life, and placing a continued and sustained financial burden on patients and their families.[5 6] Unfortunately, while drugs are the mainstay of treatment, they do not reduce the excess decline in lung function that is common in CRDs.[7] Poor availability and affordability of drugs for CRDs in LMICs limit their impact, and effective non-pharmacological treatments could help mitigate this deficit.[8]

The World Health Organisation (WHO) recognised the global burden of non-communicable diseases through its 'Rehabilitation 2030: A Call for Action' initiative.[9] Rehabilitation is a set of interventions to address limitations in everyday physical, mental and social functioning due to ageing, or a specific health condition such as chronic disease or injury.[10] For individuals with CRDs, pulmonary rehabilitation (PR) is one of the core treatment modalities. This is a programme of exercise training, education and behaviour management, designed to improve the physical and psychological condition of people with CRDs and to promote the long-term adherence of health-enhancing behaviours.[11] Notably, in people with COPD, PR is supported by high-quality evidence of improvement in symptoms (dyspnoea, fatigue, anxiety, depression), exercise tolerance and overall health-related quality of life.[12] There is also evidence supporting PR for other CRDs including asthma,[13] p-TBLD[14] and bronchiectasis.[15] In high-income countries, PR significantly reduces the direct costs of COPD by decreasing healthcare system usage, particularly unplanned hospital admissions.[16] In LMICs, low-cost modifications can reduce the costs of specialist equipment, potentially allowing wider access and increased feasibility.[17 18]

Current PR evidence is predominantly based on studies from high-income countries.[19] Clinical PR services are not widely available in LMICs,[20] where significant modifications may be required due to differences in resources, awareness, culture, healthcare configuration and target disease epidemiology.[21] A systematic review of PR in Southern Africa demonstrated few trials, generally low-quality evidence for efficacy, and no published data from Malawi.[3] Our proposed trial aims to investigate the feasibility and acceptability of implementing PR in Malawi, with the following specific objectives:

1. To codesign, with service users and stakeholders, a locally appropriate PR programme for adult patients with functionally limiting CRDs in Malawi.
2. To examine feasibility (ie, participants' recruitment, retention, engagement, protocol adherence) and acceptability of the programme. This trial adapts the definition of 'acceptability' by Sekhon *et al*[22]; the extent to which participants in this trial consider the PR programme to be appropriate, based on their experienced cognitive and emotional responses to the intervention. Elements of acceptability will include the participants' attitudes, burden, perceived effectiveness, ethicality, intervention coherence, opportunity costs and self-efficacy.[22]
3. To describe changes in participants' lung function, exercise capacity and health status following their completion of the PR programme.

Findings of this trial will inform the design of a multicentre randomised controlled trial of PR in Malawi to help address the broader question, 'In low-resource settings (ie, like Malawi, settings with suboptimal healthcare service delivery for people with CRDs, underdeveloped healthcare infrastructure including lack of PR specialist equipment, paucity of PR knowledge and expertise, and shortage of human resources for health including rehabilitation professionals),[23] what is the feasibility and effectiveness of PR?'

## METHODS AND ANALYSIS

The trial will be conducted, analysed and reported according to the Standard Protocol Items: Recommendations for Interventional Trials statement.[24] The trial is prospectively registered on the ISRCTN website (https://doi.org/10.1186/ISRCTN13836793).

### Trial design

The proposed feasibility trial will use a single-centre mixed-methods pre–post single-arm design and will be conducted in three phases as follows:

► Phase 1: pre-trial qualitative work to inform the modifications required to make PR specific to the Malawi context.
► Phase 2: trial of the 6-week PR programme.
► Phase 3: post-trial quantitative and qualitative evaluation to determine the feasibility and acceptability of PR among participants, followed by identification of future modifications to the PR programme by the PR delivery team.

## Trial place

The trial will be conducted at a single site, namely Queen Elizabeth Central Hospital (QECH), Blantyre, Malawi.

## Trial period

Phase 1 is expected to run from October 2021 to November 2021. Phase 2 is expected to run from November 2021 to February 2022. Phase 3 is expected to run from February 2022 to March 2022.

## Target population, sample size, recruitment and eligibility

The trial population will comprise patients presenting with functionally limiting CRDs such as p-TBLD, COPD, asthma and bronchiectasis, at QECH. Ten patients will be recruited, and these will be invited to participate in both phase 1 and phase 2 of the study. In phase 2 (the trial phase), they will be divided into groups of five per PR class or session. This sample size was chosen pragmatically based on previous PR studies of similar sample sizes[25 26] (ie, 7 and 12 participants, resepctively). Patients will be consecutively sampled and recruited from the chest clinic and medical wards at QECH. Inclusion criteria are: aged 18 years or older; a spirometry confirmed diagnosis of any CRD such as COPD, p-TBLD, asthma and bronchiectasis; functional limitation due to breathlessness reaching a score of ≥2 on the modified Medical Research Council (mMRC) dyspnoea scale.[27] Exclusion criteria are those who do not meet the inclusion criteria above; have acute or unstable conditions and medical complications (eg, acute myocardial infarction, uncontrolled asthma, syncope, thrombosis, pulmonary oedema, uncontrolled arrhythmias causing symptoms or haemodynamic compromise, acute respiratory failure, mental impairment leading to inability to cooperate, etc) or any other condition that would compromise participation in the rehabilitation programme; have an active infection including TB and COVID-19; have respiratory disease which is thought primarily to originate from COVID-19; are unable to provide informed consent.

## Data collection

### Phase 1

Face-to-face semistructured in-depth interviews with eligible participants will explore their experiences of living with CRD, including functional limitations, and identify potential enablers and barriers to participating in or adhering to PR. These interviews will be conducted in a quiet, private room. Prior to each interview, the reseachers will explain the study to each participant using a detailed participant information sheet (online supplemental file 1), written in plain Chichewa (most used language in Malawi) and English, so that each participant can make an informed decision to participate or not. The researchers will read the participant information sheet loudly and verbatim at a moderate pace. This will be followed by a question and answer session between the participant and reseachers, respectively, to ensure that all the participant's questions and concerns about the study

are satisfactorily addressed by the reseachers. Then, the participant will be asked to give their written informed and autonomous consent to participate or not. Interviews are expected to last for up to 1 hour and will be audio-recorded with prior consent of each participant. A topic guide (online supplemental file 2) will direct the content of the interview and will include participants' experiences with breathing difficulties, anticipated enabling and limiting factors to undertaking a PR programme, attitudes and views on different elements or components of the programme, and suggestions for making the programme culturally acceptable to them.

### Phase 2

As already mentioned, phase 1 participants will be invited to participate in phase 2 of the study. A case report form (CRF) will collect the following data from phase 2 participants (ie, through self-reported data and measurements) and their medical files:

► Demographic variables including age, sex, ethnic group, highest education level and employment status.

► Lung health variables including potential risk factors for lung disease (these will be assessed by adapting IMPALA questionnaires[28] available at https://github.com/jipp3r/IMPALA_QuestionSet, for example, smoking status (pack years) and biomass fuel exposure), primary respiratory diagnosis, time since diagnosis in years, number of CRD exacerbations and hospitalisations within the last 12 months, spirometric measurements (prebronchodilator and postbronchodilator forced expiratory volume in one second ($FEV_1$), forced vital capacity (FVC) and $FEV_1$/FVC ratio); comorbidities and treatments.

► Anthropometric measurements of height (using a Holtain stadiometer), weight (using calibrated weight scale) and body mass index (BMI).

► Health status variables including participants' perceived respiratory disability due to dyspnoea (using the mMRC dyspnoea scale[29]), health-related quality of life (using the COPD Assessment Test[30]), subjective experience of fatigue (using the Checklist Individual Strength fatigue subscale[31]) and psychological well-being (using the Hospital Anxiety and Depression Scale[32])—these tools will be translated and administered to participants in a standard manner in Chichewa, with the caveat being that they have not been cross-culturally validated in Malawi.

► Physical fitness variables including exercise tolerance (using the Incremental Shuttle Walk Test[33]), with before-test and after-test measurements of oxygen saturation ($SpO_2$) and heart rate (using a pulse oximeter), blood pressure if not recently documented in the participant's medical file (using a sphygmomanometer), Borg rating of breathlessness[34] and Borg rate of perceived exertion.[34] We will also assess lower extremity muscular strength (using five-repetition

sit-to-stand test[35]) and hamstring flexibility (using the chair sit and reach test[36]).

Using combination of researcher observation, interaction with the participants, field notes and informal interviews, we will also assess the feasibility of running the PR programme in this study in the following areas: participants' recruitment (ie, availability of eligible participants and their referral to our PR programme); retention (ie, number and reasons of participant dropouts); engagement (ie, participants' enthusiasm with the intervention); and protocol adherence to the exercise regimen at both home and hospital.

## Intervention description

Interview data from phase 1 will be used to culturally adapt the PR programme. We initially propose a 6-week, twice-weekly, supervised centre-based PR programme, with an additional weekly home-based non-supervised session. The programme will be delivered in a group format and will consist of about an hour of exercise training and another hour of health education. The PR sessions or classes will be run in the Physiotherapy Department at QECH. Participants will be provided with refreshments (ie, during rest periods of exercise sessions) and reimbursed for transport at rates mandated by the local research ethics committee. The initial intensity of the exercise regime will be tailored to each participant's exercise capacity (ie, using Incremental Shuttle Walk Test performance results for individual participant, that is, 50%–75% of their peak speed achieved on the Incremental Shuttle Walk Test[37]), and gradually increased according to each patient's improvement over the course of PR (eg, as they are able to do more repetitions of an exercise than they were on the previous session).[38]

At each session, participants will complete a set of exercises designed to improve aerobic fitness, muscle strength, endurance and flexibility. Established principles of exercise training will be used during exercise prescription.[39] Each participant will also be given an information sheet containing illustrations and instructions written in Chichewa for their weekly home-based exercise session. Participants will be requested to keep a standardised exercise and physical activity diary at home, which will have been pilot-tested a priori. The PR programme will not interfere with the participants' routine medical care from the primary care and/or referring teams.

The education component will be delivered to participants and their families or caregivers, which address: patients as active participants in their healthcare; understanding of the physical and psychological changes that occur with chronic illness, and coping strategies for these.[40] Education will be delivered either in group or individual sessions depending on the sensitivity of the topic and insights from phase 1. Table 1 below lists some of the topics that may be covered.[41]

| Table 1 | Education topics |
| --- | --- |
| **Topics that may be suitable for group sessions** | **Topics that may require individual consultation** |
| Information about the lungs, lung diseases and respiratory medications | Smoking cessation |
| Exacerbations: prevention and management including action plans | Inhaler technique |
| Benefits of physical activity and exercise | Nutrition |
| Causes and management of breathlessness | Oxygen therapy |
| Coping with chronic lung disease, management of depression, anxiety and panic attacks | Instruction in airway clearance techniques |
| Communication with health professionals | Issues related to travelling with lung disease |
| Community resources including home care options | Sexual intimacy |
| | Continence |
| | Advanced care planning and end-of-life decision-making |

## Phase 3

Repeat assessments of participants' lung function, anthropometry (weight, height, BMI), submaximal exercise tolerance and health status will be conducted. Post-trial interviews with participants will also be conducted using a topic guide (online supplemental file 3), to assess the acceptability of the programme to participants. All participants who either complete or do not complete the 6-week PR programme will be invited to participate in these interviews. They will be asked for what went well and what might be improved in the programme. The interviews will be conducted in a private room at QECH (or at participant's home if they are unable to travel to QECH due to illness, etc), and are expected to last for up to 1 hour and will be audio-recorded with prior consent.

## Patient and public involvement

As described in phase 1 and 2 sections, patients will be involved in the design of a culturally appropriate PR programme for this trial using their data collected through in-depth interviews prior to PR trial phase. After the trial, we will recruit a patient and public involvement (PPI) representative (patient expert) from the trial participants' group who can help, in a formal role, with the dissemination of trial findings and PPI in future studies. Currently, the protocol has received both internal and external expert input.

## Data management

The CRFs will be printed out so that participants' data collected by these will be stored in physical form. All CRFs with participants' data will be stored securely in a locked

cabinet in the Physiotherapy Department at QECH. These will later be anonymised and digitally archived on a public online database similar to IMPALA's peer-reviewed 'Questionnaires for Lung Health across the Life Course' (https://github.com/jipp3r/IMPALA_QuestionSet),[42] as a harmonised and shared data collection system of demographics, baseline and PR outcomes for patients undergoing PR in Malawi. The management of the system will be supported by the Data Management Support Unit at the Malawi-Liverpool-Wellcome Trust Clinical Research Programme. Likewise, interview data will be anonymised at the time of translation and transcription (ie, participants' names and dates will be removed). All anonymised transcripts will be stored for a maximum of 5 years on a password-encrypted computer at Malawi-Liverpool-Wellcome Trust Clinical Research Programme, after which they will be permanently deleted.

### Quantitative data analysis and presentation
Quantitative data will be imported to IBM SPSS Statistics V.27 for descriptive analysis. The average pre–post PR outcome differences or changes in clinical outcomes will be represented as relative and absolute changes. Non-parametric statistical testing (Wilcoxon Signed-Rank Test[43]) will be used to explore differences, with the caveat that this feasibility trial is not designed a priori for statistical power to detect differences or changes in clinical outcomes. In addition, as the trial lacks a control group with which to compare any recorded difference in clinical outcomes, we will be unable to attribute any pre–post PR difference or change in clinical outcomes to PR. However, where appropriate, the average changes in clinical outcomes will be discussed in relation to published minimal clinically important differences.[44] Results from quantitative data analysis will be presented in tables and/or graphs, with mean (SD), median (IQR) or frequency (percentage) used to report summary data.

### Qualitative data analysis and presentation
Anonymised interview transcripts will be imported into QSR NVivo software platform for thematic analysis.[45] The analysis will be performed collaboratively between researchers to encourage a breadth and depth of analysis from different perspectives and viewpoints. Initially, the researchers will collaborate to develop a preliminary coding framework. Data coding will be performed by the researchers separately, using an iterative inductive approach[45] across all qualitative data. In summary, the researchers will broadly follow the steps outlined by Braun and Clarke,[45] including going through a process of familiarisation with the data by reading and rereading transcripts while making reflective notes on the literal content, looking closely at words used by participants, interpreting what the data meant by assigning initial codes or classifications to segments of text, and exploring relationships between these classifications and developing core themes. Results from qualitative data analysis will be presented in narrative form based on key themes established during the analysis, supplemented by participants' quotations.

### Adverse events
For the purposes of this work, an adverse event is defined as any unfavourable or unintended sign or response associated with participating in the PR programme of this trial. Due to anticipated thorough participant assessment (including risk assessment) and individualised nature of PR in this trial, adverse events are not anticipated. However, if such events do occur, the programme will be stopped immediately for the affected participant so that they can be given appropriate management, which may include referral to the medical team. All adverse events will be registered on the adverse event log and CRF. Serious adverse events will be reported to the Clinical Research Support Unit at Malawi-Liverpool-Wellcome Trust Clinical Research Programme, College of Medicine Research and Ethics Committee (COMREC), and University of Leicester Research Ethics Committee, for guidance where required.

## ETHICS AND DISSEMINATION OF THE RESULTS
Ethical approval for this trial was obtained from University of Malawi COMREC, Blantyre, Malawi (protocol number: P.07/19/2752), and University of Leicester Research Ethics Committee, Leicester, United Kingdom (ethics reference: 31574). The results of the trial will be disseminated through oral presentations at local and international scientific conferences or seminars and publication in a peer-reviewed journal. We will also engage the participants who complete the PR trial and the Science Communication Department at Malawi-Liverpool-Wellcome Trust Clinical Research Programme to organise community outreach activities within Blantyre to educate communities about CRDs and PR. We will also broadcast our trial results through national radio station programmes such as the weekly 'Thanzi la Onse' (Health of All) programme by Times Radio Malawi. We will formally present our trial results to Blantyre District Health Office and Malawi Ministry of Health.

**Author affiliations**
[1]Lung Health Research Group, Malawi-Liverpool-Wellcome Trust Clinical Research Programme, Blantyre, Malawi
[2]UCL GOS Institute of Child Health, University College London, London, UK
[3]Physiotherapy Department, Queen Elizabeth Central Hospital, Blantyre, Southern Region, Malawi
[4]Clinical Sciences, Liverpool School of Tropical Medicine, Liverpool, UK
[5]Department of Respiratory Sciences, University of Leicester, Leicester, UK
[6]Centre for Exercise and Rehabilitation Science, NIHR Leicester Biomedical Research Centre Respiratory Diseases, Leicester, UK
[7]Medicine Department, Queen Elizabeth Central Hospital, Blantyre, Southern Region, Malawi
[8]Department of Rehabilitation Sciences, Kamuzu University of Health Sciences, Blantyre, Malawi
[9]Department of Economics, University of Sheffield, Sheffield, UK
[10]NIHR Leicester Biomedical Research Centre, University Hospitals of Leicester NHS Trust, Leicester, UK

 

**Contributors** FMB drafted the manuscript based on the original study protocol conceived and designed by FMB, JR and EC. FMB sent the draft manuscript to JR, EC, TM, RN, MWO, CF, MM, LM, WL, LW, IP, ZKY, ZA, MK, FL, HS, AB, RCF, MS, JAM, AM and SJS for their critical reviews for important intellectual content. FMB revised the draft accordingly before submission to BMJ Open. All authors approved the final revised draft to be submitted to BMJ Open and are accountable for all aspects of the work.

**Funding** This work was supported by the National Institute for Health Research (NIHR) (using the UK's Official Development Assistance (ODA) Funding) and Wellcome [221465/Z/20/Z] under the NIHR-Wellcome Partnership for Global Health Research; the Royal Society of Tropical Medicine and Hygiene (RSTMH) 2019 Small Grants programme (funded by NIHR; award/grant number N/A); the Academy of Medical Sciences Global Challenges Research Fund (GCRF) Networking Grant Scheme (GCRFNGR5\1242); and the NIHR- Global Health Research Group on Respiratory Rehabilitation (Global RECHARGE) (17/63/20) using UK aid from the UK Government to support global health research.

**Disclaimer** The views expressed are those of the authors and not necessarily those of the Wellcome, RSTMH, NIHR or the Department of Health and Social Care.

**Competing interests** None declared.

**Patient consent for publication** Not applicable.

**Provenance and peer review** Not commissioned; externally peer reviewed.

**ORCID iDs**
Fanuel Meckson Bickton http://orcid.org/0000-0002-0925-909X
Laura Wilde http://orcid.org/0000-0002-1404-6304
Zainab K Yusuf http://orcid.org/0000-0001-7859-5102
Mark Orme http://orcid.org/0000-0003-4678-6574

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
