## [Reviewer comments · BMJ Open]

ARTICLE DETAILS

TITLE (PROVISIONAL)	Protocol for a single-centre mixed-methods pre-post single-arm feasibility trial of a culturally appropriate six-week pulmonary rehabilitation programme among adults with functionally limiting chronic respiratory diseases in Malawi
AUTHORS	Bickton, Fanuel; Mankhokwe, Talumba; Nightingale, Rebecca; Fombe, Cashon; Mitengo, Martha; Mwahimba, Langsfield; Lipita, Wilfred; Wilde, Laura; Pina, Ilaria; Yusuf, Zainab; Ahmed, Zahira; Kamponda, Martin; Limbani, Felix; Shannon, Harriet; Chisati, Enoch; Barton, Andy; Free, Robert; Steiner, Michael; Matheson, Jesse; Manise, Adrian; Singh, Sally; Rylance, Jamie; Orme, Mark

VERSION 1 – REVIEW

REVIEWER	Heine, Martin Stellenbosch University, Institute of Sports and Exercise Medicine
REVIEW RETURNED	27-Sep-2021

GENERAL COMMENTS	Dear Mr. Bickton, Congratulations on an excellent written and thought-out study on the feasibility of pulmonary rehabilitation in a Malawian hospital setting. I found the manuscript excellently written yet have a few questions and suggestions. Page 5, Line 54; Typo (“broder”) Page 5, Line 52 – 56. To appreciate the external validity of the multi-centre study to a wider African context, one would need to have a better understanding of the complex setting in which the multi-centre trial will take place. Our group recently developed a synthesizes of various setting-related features that you could consider in this respect: https://gh.bmj.com/content/6/6/e005190 Page 6, line 54; I would suggest making it a deliberate effort to include a minimum number of participants in phase 2 that were not part of phase 1. One could argue that participation in phase 1 will bias the outcome of feasibility in phase 2. Page 7, line 10; I would suggest making the diagnoses under consideration explicit. Page 7, line 17 – 20; Multimorbidity (hypertension, diabetes) is an important aspect of chronic disease management in low-resource settings. However, given the person-centred nature of rehabilitation, there are few comorbidities that should warrant exclusion. Please clarify.
---

	Page 7, line 58; What is meant with ethnicity, and how is this relevant to the feasibility of the program. If ethnicity is an umbrella for other relevant factors, I'd suggest including those rather. Furthermore, given the centre-based nature of this program, I would consider other features like household income and access to transport, as these may affect uptake / adherence. Page 7/8, data collection; please clarify, will access be obtained to the patient' medical file? Based on our experiences in Cape Town, South Africa I would suggest doing so (may depend on the setting of course), as this greatly increased the quality of data collection through triangulation of self-report data and the medical record (e.g., for comorbidity). Page 8, Health Status; please clarify in what language these measures will be provided? An English-only set-up may bias feasibility outcomes and quality of data collection; and I see that instructions for the intervention are provided in Chichewa? Furthermore, what is known on the cross-cultural validity of these self-report measures in the Malawian context (see https://trialsjournal.biomedcentral.com/articles/10.1186/s13063-021-05328-z) Page 8, Health Status; based on experiences in other low-resource settings, I would suggest including a measure of health-literacy, as this may inform aspects like perceived value of PR, quality of data collection, etc. Page 8, Lung Health; how about non-smoking tobacco products? Biomass full exposure, how is this assessed? Page 8, line 33; resting heart rate? Standardization? Page 8, line 51 – 53; How does remuneration of participation in the PR program (i.e. the actual PR sessions) affect the external validity of the study findings when considering implementation of PR in the existing health systems? Our experiences across low-resource settings across South Africa have shown that such remuneration can be a key driver for participation, particularly in settings with low socio-economic quality of life. We therefore opted to only reimburse participants for “research visits” (e.g., interviews, assessments) and not for participating in the rehabilitation sessions. Please clarify. Page 8, line 41; consider adding sub-heading for describing the intervention. The details on how the exercise regime is standardised could be expanded on (i.e., how is the ISW used, and how is “improvement” defined, in order for gradual increase to take place; what is gradual etc?) Page 9, line 16; Please make sure that the exercise diary is pilot tested a priory. We had major issues in our study (Heine et al. 2019, BMJ Open) in this respect, where the therapist ended up retrospectively completing the diary on behalf of the patient, introducing all sorts of bias. Page 9, line 22; Great that families and caregivers are involved in the education component. Page 10, phase 3; Please clarify why only patients that completed the PR program are invited to participate in phase 3? Would it now
--	--

	provide a wealth of information as to reasons for non-completion, feasibility, implementation etc specifically from those that did not complete the program? Please clarify Page 10, line 52; will the quantitative data be linked, through the identified, to the qualitative data? Page 12, adverse events; Just as a matter of interest, in our experience, working in low-resource settings, we find that numerous clinical aspects may arise (e.g. hypertension, uncontrolled diabetes) during the risk assessment. This obviously depends on the quality of care in the specific context, and the level to which the risk assessment provides a more thorough clinical assessment relative to usual care. However, technically, these could be considered adverse events not related to the treatment yet may affect feasibility of the program when implemented in a real-world setting where such thorough risk assessment is absent. Supplemental 1: The description of PR seems quite high-level; I find it difficult to gauge to what extent the interviewees would be able to engage with this description sufficiently (see also my suggestion to add health literacy as a measure of interest). Maybe there are patients that already participated in a local PR program that might provide early feedback on this description?
--	---

VERSION 1 – AUTHOR RESPONSE

Reviewer: 1

Dr. Martin Heine, Stellenbosch University

Comments to the Author:

Dear Mr. Bickton,

Congratulations on an excellent written and thought-out study on the feasibility of pulmonary rehabilitation in a Malawian hospital setting. I found the manuscript excellently written yet have a few questions and suggestions.

1. Page 5, Line 54; Typo (“broder”)

Author response: The typo has been corrected as “broader”.

2. Page 5, Line 52 – 56. To appreciate the external validity of the multi-centre study to a wider African context, one would need to have a better understanding of the complex setting in which the multi-centre trial will take place. Our group recently developed a synthesizes of various setting-related features that you could consider in this respect: <https://gh.bmj.com/content/6/6/e005190>

Author response: We agree and thank you for sharing with us a reference for an important work that unravelled “low-resource settings” (LRSs). We have now used this term instead of the umbrella term “LMICs” which does not grasp the complex interplay of resource constraints of “LRSs” and because of the weak association between the two. To further put it into context, as suggested by authors of the cited work above, we have also given examples of some of the key resource constraints facing Malawi as follows:

Findings of this trial will inform the design of a multi-centre randomized controlled trial of PR in Malawi to help address the broader question, “In low-resource settings (i.e., like Malawi, settings with suboptimal healthcare service delivery for people with CRDs, underdeveloped healthcare infrastructure including lack of PR specialist equipment, paucity of PR knowledge and expertise, and shortage of human resources for health including rehabilitation professionals (van Zyl et al, 2021)), what is the feasibility and effectiveness of PR?”

3. Page 6, line 54; I would suggest making it a deliberate effort to include a minimum number of participants in phase 2 that were not part of phase 1. One could argue that participation in phase 1 will bias the outcome of feasibility in phase 2.

Author response: Thank you for this thoughtful insight. Firstly, please note that due to some project funding constraints that have recently arisen, we will recruit only 10 participants, which is the original sample size we had proposed in the original protocol for the study in 2018. An amendment on the same will soon be submitted to the ethics committee for approval. With the 10 participants, we hope to finalise phase 1 (qualitative phase) of the study this month of November, having already recruited and collected data from nine participants. All the nine participants recruited and interviewed so far have expressed interest to participate in phase 2 (intervention phase) of the study and our new plan (to be included in the amendment to be submit for ethical approval) is to recruit only these phase 1 participants in phase 2 (after final eligibility checks). The reasons for retaining phase 1 participants in phase 2 are in line with the heuristic framework for the cultural adaptation of interventions (<https://onlinelibrary.wiley.com/doi/abs/10.1111/j.1468-2850.2006.00043.x>) as follows:

- The main aim of the pre-trial qualitative work is to adapt the planned intervention culturally and individually to participants as evidence shows that a lack of attention to cultural and individual factors in the delivery of PR may be a barrier to its uptake by patients. So, the resulting PR programme from the pre-trial qualitative work will reflect the cultural and individual needs of 10 participants we will have interviewed. Therefore, it would be surprising to deliver it to new patients (i.e., those that did not participate in the intervention design in Phase 1) as that would defeat the very purpose of Phase 1, since this new cohort would require different cultural and individual adaptations. In other words, Phase 2 will test an intervention programme on patients on whom the programme was adapted.

- Being a “pre-post” trial, we want to compare the participants’ “pre” data with their “post” data. The former includes pre-trial qualitative data while the latter includes post-trial qualitative data and more accurate comparisons between the two qualitative data sets can only be possible if we have both qualitative data sets for each participant in the first place.

- In line with the final stage of the framework (adaptation refinement), the post-qualitative interviews will also inform further modifications to the tested programme which will now be ready for further testing in a multi-centre RCT to further improve its feasibility on wider scale. In other words, the intervention feasibility established by this small and one-centred study will not be regarded as definitive for Malawi, but will have at least improved our knowledge of feasibility in this setting and given us a head start for the RCT.

However, for patient-reported outcomes from the interviews and HRQoL measures, a reporting bias among participants is anticipated, a common challenge with non-pharmacologic treatment (rehabilitation) studies like ours where we will not blind participants as this is often hard to achieve with such studies (https://journals.lww.com/ajpmr/Abstract/2020/03000/Blinded_or_Nonblinded_Randomized_Controlled_Trials.2.aspx). This is a limitation we will transparently report in the manuscript that will result from the study.

4. Page 7, line 10; I would suggest making the diagnoses under consideration explicit.

Author response: We have specified examples of common “CRDs” under consideration, namely: COPD, post-tuberculosis lung disease, asthma and bronchiectasis.

5. Page 7, line 17 – 20; Multimorbidity (hypertension, diabetes) is an important aspect of chronic disease management in low-resource settings. However, given the person-centred nature of rehabilitation, there are few comorbidities that should warrant exclusion. Please clarify.

Author response: We agree, and it is almost impossible in our setting to find an adult patient with a single chronic disease (i.e., not uncommon to find those with a CRD diagnosis who also has hypertension, etc). We, therefore, anticipate some of our participants to have multimorbidity. PR is recommended and primarily responsive to people with CRDs, and we aim to include patients with functionally limiting CRDs and these may include those with other comorbidities if those commodities are medically controlled (i.e., stable, non-acute). All participants will undergo final doctor and physiotherapist checks to confirm eligibility and there will also be close monitoring of each participant during the intervention to ensure safety (also see “adverse events” section). But, yes, there are some absolute contraindications to PR, and we have given examples of these (i.e., acute, or unstable conditions and medical complications such as acute myocardial infarction, uncontrolled asthma, syncope, thrombosis, pulmonary oedema, uncontrolled arrhythmias causing symptoms or haemodynamic compromise, acute respiratory failure, and mental impairment leading to inability to cooperate).

6. Page 7, line 58; What is meant with ethnicity, and how is this relevant to the feasibility of the program. If ethnicity is an umbrella for other relevant factors, I'd suggest including those rather. Furthermore, given the centre-based nature of this program, I would consider other features like household income and access to transport, as these may affect uptake / adherence.

Author response: By “ethnicity”, we meant each participant’s “ethnic group” and we have used the latter term. Malawi has at least 10 major ethnic groups (the Chewa, Nyanja, Lomwe, Yao, Tumbuka, Sena, Tonga, Ngoni, Ngoni, Ngonde, Lambya) each with distinct cultural values and beliefs. A people’s culture is a major health determinant and evidence suggests that lack of attention to cultural factors in the delivery of PR may be a barrier to its uptake by indigenous ethnic groups (<https://pubmed.ncbi.nlm.nih.gov/27022255/>). By documenting the ethnic group of each of our participants, we will be able to design and deliver an indigenous-led or culturally responsive PR programme for our participants.

In agreement with the reviewer, and echoing what our recruited participants have said, transport to the PR centre is a challenge due to long distance to travel and/or lack of transport money. This is also commonly reported PR uptake barrier worldwide. To remove this barrier, we proposed to provide our participant with transport money. We will provide transport money to each participant regardless of their household income.

7. Page 7/8, data collection; please clarify, will access be obtained to the patient’ medical file? Based on our experiences in Cape Town, South Africa I would suggest doing so (may depend on the setting of course), as this greatly increased the quality of data collection through triangulation of self-report data and the medical record (e.g., for comorbidity).

Author response: Yes, some data will be collected from the participants’ medical files and have revised as follows: “A case report form will collect the following data from phase 2 participants (i.e., through self-reported data and measurements) and their medical files...”

8. Page 8, Health Status; please clarify in what language these measures will be provided? An English-only set-up may bias feasibility outcomes and quality of data collection; and I see that instructions for the intervention are provided in Chichewa? Furthermore, what is known on the cross-cultural validity of these self-report measures in the Malawian context (see <https://trialsjournal.biomedcentral.com/articles/10.1186/s13063-021-05328-z>)

Author response: Firstly, we agree with the reviewer that English-only health status measures may be problematic to administer to the Malawi population where English is not a native language, in addition to those measures (mMRC dyspnoea scale, CAT, CIS- Fatigue scale, and HADS) having been developed and validated in developed countries of Global North, except for CAT which has been once validated in some populations of North Africa and Middle East regions (<https://www.sciencedirect.com/science/article/pii/S0954611112700173>) but none in the southern Africa region (to our best knowledge) including in Malawi, although a single case has been reported where the English-only mMRC dyspnoea scale, CAT, and CIS-Fatigue scale have recently been used successfully in a pulmonary rehabilitation programme of one Malawian patient in the post-acute COVID-19 phase (https://journals.lww.com/ajpmr/Fulltext/2021/03000/An_Improvised_Pulmonary_Telerehabilitation_Program.2.aspx). Noteworthy, our study participants are coming from different ethnic groups which have distinct ethnic languages as are the researchers. However, all the participants and researchers will be commonly communicating in Chichewa, and we translated all the health status measures above into a Chichewa. However, none of the tools above has been validated in any of the Malawian languages including Chichewa and we acknowledge this lack of cross-cultural validation as another limitation of our study, to be transparently reported after study completion so that our findings can be interpreted with caution. However, we also do acknowledge the recent tremendous of the International Multidisciplinary Programme to Address Lung Health and TB in Africa (IMPALA) project for designing a set of tools/questionnaires with improved relevance to southern Africa for the assessment of chronic lung disease and risk factors (<https://www.mdpi.com/1660-4601/15/8/1615>). These questionnaires have not yet been fully field-evaluated but were already translated and validated into Chichewa, and we will adapt them in our CRF's patient history-history taking section as well as in the participant interviews of phase 1 as probing questions.

9. Page 8, Health Status; based on experiences in other low-resource settings, I would suggest including a measure of health-literacy, as this may inform aspects like perceived value of PR, quality of data collection, etc.

Author response: We agree with the reviewer that health literacy (HL) is an important measure, with evidence suggesting that people with COPD who have low HL have higher risk of comorbidities, feeling helplessness, anxious or depressed, hospital admission, adherence to medications, visits to emergency departments, and all-cause mortality (<https://www.sciencedirect.com/science/article/pii/S0147956321000388#bib0032>). A study in the Netherlands found that low HL was common among patients entering PR and their loved ones, although outcomes (including drop rate from the PR programme) were comparable between patients with low and adequate HL (<https://pubmed.ncbi.nlm.nih.gov/33797458/>). It is reported that developing countries like Malawi bear the largest burden of limited HL. Unfortunately, the current HL measures are of limited use to these countries as they have largely emanated from developed countries, reflecting the characteristics of their economies, populations, and health systems. Specifically, there exists no validated HL measure for Malawian population with CRDs, let alone within the PR context. Therefore, we will not formally measure HL as doing that would be too involving and resource-demanding for the present minimalistic feasibility study, including ensuring that the content of the developed or adapted measure is first contextualized to the Malawi population with CRDs (cultural and ethnocentric relevance, etc) (https://www.tandfonline.com/doi/full/10.1080/17538068.2016.1147742?casa_token=UBWgqZZnKLU)

AAAAA:3KAWXy7TpWYzJcmMDmGHO2XW8fll6DxMTuY9Q9LEwDcaUsCBmkVqjB16YH5X7W9dU CKdDXnQqm3h6zc). However, the PR programme to be delivered in this study will include the education component which will likely improve HL among participants will include basic topics such as pathophysiology of participants' various CRDs, exercise benefits and self-management. Post-trial interviews will collect participant-reported data on any changes following their completion of the PR programme, including changes in the knowledge of their disease and self-management. These will be qualitatively reported in form of participant's' quotes. But we hope to comprehensively study HL in a similar population of a future study.

10. Page 8, Lung Health; how about non-smoking tobacco products? Biomass full exposure, how is this assessed?

Author answer: As previously mentioned, our CRF will adapt questions from the IMPALA "questionnaires for lung health in Africa across the life course" (<https://pubmed.ncbi.nlm.nih.gov/30065166/>) which were specifically designed for survey use in southern Africa for assessment of respiratory disease and risk factors. (a) Yes, some of the questions our CRF will adopt/adapt from these IMPALA questionnaires will be about participants' exposure to non-smoking tobacco products such as fur from animals (e.g., we will ask the participant if they have contact with animals where you live), dust exposure (e.g., we will ask the participant if their home is close to a busy dusty major road that is used by lorries and/or regular buses/minibuses), exposure to aerosols or sprays at home (e.g., we will ask participants if they regularly use insecticide, deodorant, cleaning spray and any others), and ask participants if they burn mosquito coils at home, etc. (b) as for the assessment of biomass "fuel" exposure, we will ask participants the following questions adopted from the IMPALA questionnaires: "What is the main type of fuel or energy source do you use at home e.g., for cooking, heating, lighting, etc.? What type of ventilation, if any, is present in your indoor cooking area?" The IMPALA questionnaires are available on an online platform (https://github.com/jipp3r/IMPALA_QuestionSet) for further adaptation and use by research teams across the region.

11. Page 8, line 33; resting heart rate? Standardization?

Author response: Yes, as per recommendations on how to carry out a field walking test in CRD (<https://www.ncbi.nlm.nih.gov/pmc/articles/PMC4487379/>) – in this case, incremental shuttle walk test – participants' heart rate (like SpO₂) will be continuously measured at rest (i.e., resting heart rate), during and immediately after the test, and after 15 minutes of resting (to check for stability of resting heart rate). It will be measured using a standard pulse oximeter. We have highlighted "before- and after-test measurements of..." which we had included in the paragraph, with the "before-test measurement" implying resting heart rate.

12. Page 8, line 51 – 53; How does remuneration of participation in the PR program (i.e., the actual PR sessions) affect the external validity of the study findings when considering implementation of PR in the existing health systems? Our experiences across low-resource settings across South Africa have shown that such remuneration can be a key driver for participation, particularly in settings with low socio-economic quality of life. We therefore opted to only reimburse participants for "research visits" (e.g., interviews, assessments) and not for participating in the rehabilitation sessions. Please clarify.

Author response: We agree with the reviewer; we will reimburse participants for transport only on each research visit, as transport is a frequently reported healthcare access barrier including in our setting as previously mentioned.

13. Page 8, line 41; consider adding sub-heading for describing the intervention. The details on how the exercise regime is standardised could be expanded on (i.e., how is the ISW used, and how is “improvement” defined, in order for gradual increase to take place; what is gradual etc?)

Author response: We have added the subheading “Intervention description” as per the reviewer’s suggestion. We have clarified that: “The initial intensity of the exercise regime will be tailored to each participant’s exercise capacity (i.e., using incremental shuttle walk test performance results for individual participant, i.e., 50%–75% of their peak speed achieved on the incremental shuttle walk test – <https://oce.ovid.com/article/00002060-201207000-00006/HTML>), and gradually increased according to each patient’s improvement over the course of PR (e.g., as they are able to do more repetitions of an exercise than they were on the previous session, etc)”.

14. Page 9, line 16; Please make sure that the exercise diary is pilot tested a priori. We had major issues in our study (Heine et al. 2019, BMJ Open) in this respect, where the therapist ended up retrospectively completing the diary on behalf of the patient, introducing all sorts of bias.

Author response: We thank the reviewer for this insight; we will ensure to pilot-test our exercise diary and we have revised in the manuscript as follows: “Participants will be requested to keep a standardized exercise and physical activity diary at home, which will have been pilot-tested a priori.”

15. Page 9, line 22; Great that families and caregivers are involved in the education component.

Author response: Thank you.

16. Page 10, phase 3; Please clarify why only patients that completed the PR program are invited to participate in phase 3? Would it now provide a wealth of information as to reasons for non-completion, feasibility, implementation etc specifically from those that did not complete the program? Please clarify

Author response: Thank you for this insight. We agree that participants who did not complete the program would provide us useful information in phase 3 as to reasons for their non-completion. We will interview these too and have revised as follows in the manuscript: “All participants who either complete or do not complete the six-week PR programme will be invited to participate in these interviews”.

17. Page 10, line 52; will the quantitative data be linked, through the identified, to the qualitative data?

Author response: We assume the reviewer is asking whether we will correlate quantitative exercise test outcomes (e.g., incremental shuttle walk distance) with qualitative patient-reported outcomes (e.g., participant’s interview data on impact of CRD on their quality of life or the patient-reported outcomes collected using the health-related quality of life measures, e.g., mMRC, CAT, CIS-fatigue, and HADS), before and after PR. Well, this is beyond the objectives of the present study, but it is an interesting idea to consider in our future studies. However, in our literature review, we found one study on this in 518 patients with COPD in The Netherlands (<https://pubmed.ncbi.nlm.nih.gov/32891156/>), which reported that, overall, correlations between patient-reported outcomes (PROs) and quantitative exercise test outcomes at baseline were statistically significant. However, generally very weak, or weak correlations between the two after PR were found. The authors concluded: “... we have found that patient-reported outcomes and exercise test outcomes, although significantly correlated with each other, assess different disease features in patients with COPD. Therefore, it can be stated that relevant features from the patient’s perspective like HRQoL, anxiety, depression, and the level of care dependency are not an accurate reflection of a patient’s exercise capacity. The only exception to this seems to be dyspnea, the only PRO that

tended to imply at least moderate association with exercise test outcomes. We would like to highlight the complexity of evaluating the effectiveness of a personalized PR program, in which we note that changes in PROs and changes in exercise test outcomes correlate poorly. Indeed, improvements in exercise capacity obtained after PR do not necessarily result in alterations in PROs in patients with COPD. Individual PROs need to be supported by additional functional measurements whenever possible, in order to get a more detailed insight in the effectiveness of a PR program.”

18. Page 12, adverse events; Just as a matter of interest, in our experience, working in low-resource settings, we find that numerous clinical aspects may arise (e.g., hypertension, uncontrolled diabetes) during the risk assessment. This obviously depends on the quality of care in the specific context, and the level to which the risk assessment provides a more thorough clinical assessment relative to usual care. However, technically, these could be considered adverse events not related to the treatment yet may affect feasibility of the program when implemented in a real-world setting where such thorough risk assessment is absent.

Author response: We agree with reviewer. We will ensure to conduct a thorough risk assessment of our participants and have those that experience any adverse event properly managed, as described in the adverse events section.

19. Supplemental 1: The description of PR seems quite high-level; I find it difficult to gauge to what extent the interviewees would be able to engage with this description sufficiently (see also my suggestion to add health literacy as a measure of interest). Maybe there are patients that already participated in a local PR program that might provide early feedback on this description?

Author response: Thank you for this comment. Please note that the PR description section in that supplementary file (now supplementary file 2) is a summary of a comprehensive PR description in the participant information sheet that we have (see supplementary file 1) – we are making sure to spend enough time (20 to 25 minutes from our experience with the nine participants we have already interviewed) to explain to our participants what the study is about including PR description and allow them to ask questions. We acknowledge that fact none of our participants have prior PR experience by trying our best to use plain Chichewa and English language for PR description (and the whole study) in the participant information sheet (supplementary file 1). But from our experience with the nine participants whom we have already interviewed in phase 1, their feedback at the end of being described the intervention (and the whole study) was that they clearly understood what the intervention and the study would involve, with some already willing to undertake it before being asked to consent to participate, having previously received similar therapeutic exercises for other conditions. As such, the researchers found the PR description section in supplementary file 2 redundant and have been skipping it in the nine interviews conducted (except for the first patient interview). However, the researchers are still finding the PR description summary of supplementary file 2 useful as they may quickly refer to it (instead of the longer supplementary file 1) for a reminder of only the key elements of PR when needed by either the participant or interviewing researcher.

As for the suggestion to add health literacy as a measure of interest, we kindly refer the reviewer to our response in #9 above.

VERSION 2 – REVIEW

REVIEWER	Heine, Martin Stellenbosch University, Institute of Sports and Exercise Medicine
REVIEW RETURNED	10-Dec-2021
GENERAL COMMENTS	Well done on this excellent protocol, and all the best with this important study.